# Drivers of COVID-19 Vaccine Uptake amongst Healthcare Workers (HCWs) in Nigeria

**DOI:** 10.3390/vaccines9101162

**Published:** 2021-10-11

**Authors:** Sohail Agha, Adaobi Chine, Mathias Lalika, Samikshya Pandey, Aparna Seth, Alison Wiyeh, Alyssa Seng, Nandan Rao, Akhtar Badshah

**Affiliations:** 1Global Delivery Program, Bill & Melinda Gates Foundation, Seattle, WA 98109, USA; 2Department of Epidemiology, University of Washington, Seattle, WA 98195, USA; 3Strategic Analysis, Research & Training (START) Center, University of Washington, Seattle, WA 98195, USA; achine@uw.edu (A.C.); mlalika@uw.edu (M.L.); samikpan@uw.edu (S.P.); aseth2@uw.edu (A.S.); awiyeh@uw.edu (A.W.); akhtarb@uw.edu (A.B.); 4Department of Communication, University of Washington, Seattle, WA 98195, USA; alyssasengg@gmail.com; 5Research, Virtual Lab, Corvallis, OR 97330, USA; nandan@vlab.digital

**Keywords:** COVID-19 vaccines, health personnel, vaccine hesitancy, vaccine uptake, vaccine acceptance, willingness to get vaccinated, healthcare workers, providers

## Abstract

This study applied a behavioral lens to understand drivers of COVID-19 vaccination uptake among healthcare workers (HCWs) in Nigeria. The study used data from an online survey of Nigerian HCWs ages 18 and older conducted in July 2021. Multivariate logistic regression analyses were conducted to examine predictors of getting two doses of a COVID-19 vaccine. One-third of HCWs in our sample reported that they had gotten two doses of a COVID-19 vaccine. Motivation and ability were powerful predictors of being fully vaccinated: HCWs with high motivation and high ability had a 15-times higher odds ratio of being fully vaccinated. However, only 27% of HCWs had high motivation and high ability. This was primarily because the ability to get vaccinated was quite low among HCWs: Only 32% of HCWs reported that it was very easy to get a COVID-19 vaccination. By comparison, motivation was relatively high: 69% of HCWs reported that a COVID-19 vaccine was very important for their health. Much of the recent literature coming out of Nigeria and other LMICs focuses on increasing motivation to get a COVID-19 vaccination. Our findings highlight the urgency of making it easier for HCWs to get COVID-19 vaccinations.

## 1. Introduction

Frontline healthcare workers (HCWs) are at extremely high risk of COVID-19 infection, morbidity, and mortality. As a critical workforce in the COVID-19 response, HCWs being fully vaccinated will be key to a successful pandemic response. In addition to their own safety, HCWs exercise a disproportionate influence on the general population’s perception and acceptance of vaccination. Studies have documented the role played by HCWs in instilling confidence in the safety and efficacy of vaccines in the general population [1,2,3]. Evidence suggests there is a strong link between HCWs’ acceptance of a vaccine and its uptake in the general population.

Since early 2020, several studies have examined the acceptability of a COVID-19 vaccine amongst HCWs. They have found a wide range of levels of acceptance of a COVID-19 vaccine across different types of workers and geographies. A study conducted among HCWs in 23 referral hospitals in the Democratic Republic of the Congo (DRC), in March–April 2020, found that only 28% of HCWs were willing to accept a COVID-19 vaccine if it became available. Males and older HCWs were more willing to undergo COVID-19 vaccination. Physicians were more likely to accept a vaccine than nurses or other HCWs [3]. A study conducted among HCWs in a comprehensive, specialized, hospital in the Amhara region of Ethiopia, in February-March 2021, found that 42% of workers had a positive attitude towards COVID-19 vaccine acceptance. The Ethiopian study also found higher acceptability of a COVID-19 vaccine among physicians, as well as among older HCWs. The majority of HCWs (57%) in the study did not believe that COVID-19 vaccines worked, 62% did not believe that COVID-19 vaccines were safe, and 68% did not trust information on COVID-19 provided by the government [4].

A systematic review, comprising studies of vaccination intention in high-income countries (HICs), conducted in 2020, found similar patterns: Older HCWs were more willing to be vaccinated, physicians were more willing to be vaccinated than nurses, more educated HCWs were more willing to be vaccinated, and HCWs who had encountered or were taking care of COVID-19 patients were more willing to be vaccinated. The review concluded that addressing misinformation, especially among nurses, should be a high priority [5]. A systematic review of HCWs’ attitudes towards COVID-19 vaccination, which included seven studies from the US and two from China, found relatively low rates of vaccine acceptance [6]. The US studies, conducted from August to December 2020, found COVID-19 vaccine acceptance levels of 33%, 36%, 46%, 58%, 60%, 69%, and 77% among HCWs. The Chinese studies, conducted in February and March 2020, showed vaccine acceptance levels of 40% and 63% among HCWs. Studies in Cyprus, French Guiana, and South Africa also showed considerable variation in vaccine acceptance among HCWs [7,8,9]. Overall, the limited but growing literature on vaccine acceptance amongst HCWs suggests that concerted efforts will be needed to reduce vaccine hesitancy among HCWs in both HICs and low- and middle-income countries (LMICs).

One Nigerian study, conducted in December 2020 and January 2021, collected data on vaccine acceptance among HCWs. Less than half (49%) of sampled Nigerian HCWs were willing to get a COVID-19 vaccination. The study found substantial variation in vaccine acceptance by type of HCW: 59% of physicians compared with 38% of nurses or midwives were willing to get a COVID-19 vaccination. Nearly three quarters (74%) of HCWs reported that social media was their primary source of information about COVID-19. The study reported that disinformation circulating on digital media platforms was a barrier to a higher level of COVID-19 vaccine acceptance in Nigeria [10]. A second Nigerian study, a cross sectional survey of HCWs in four specialized hospitals, conducted in October 2020, found that slightly more than half (52%) of HCWs were willing to get a COVID-19 vaccine. Clinical HCWs were more willing to accept a COVID-19 vaccine than non-clinical HCWs (55% versus 48%). HCWs with higher than secondary education were more willing to accept a COVID-19 vaccine than those with less than secondary education [11]. In Nigeria, 6 years of secondary school follow 6 years of elementary school. A third Nigerian study, an online survey of Nigerian HCWs showed that 54% of HCWs were willing to get a COVID-19 vaccine [12].

While several studies have been recently published on Nigerian HCWs’ willingness to get a COVID-19 vaccine, these studies have not identified the drivers of COVID-19 vaccination uptake. One reason for this is that these studies have not used a behavior model to understand why preventive behaviors are adopted.

### A Behavior Model to Understand COVID-19 Vaccine Uptake among HCWs

We introduce a behavior model which has been tested in different social and behavioral contexts [13,14] in low- and middle-income countries (LMICs), including Nigeria, with the objective of identifying factors associated with the completion of COVID-19 vaccination among HCWs in Nigeria. The Fogg Behavior Model (FBM) states that behavior happens when motivation, ability, and a prompt occur in the same moment. In other words, the FBM posits that both motivation and ability must be present for behavior to occur.

The FBM can be visualized in two dimensions, with motivation on the *y*-axis and ability on the *x*-axis, as shown in Figure 1. For a specific behavior, motivation can range from high to low. Ability can also range from high to low for a particular behavior. For simplicity we say a behavior is easy to do or hard to do. The FBM proposes that a behavior happens when a person with high motivation and high ability is prompted. By contrast, a person with low motivation and low ability is not likely to adopt a behavior when prompted. Fogg considers there to be a threshold (the “action line”) above which a person with sufficient motivation and ability will adopt a behavior when prompted.

The FBM is precise in defining the components of motivation and ability: Motivation comprises anticipation (i.e., hopes or fears), sensation (i.e., pleasure or pain), and belonging (i.e., social acceptance or rejection); ability comprises time, money, physical effort, mental effort, routine, and social norms [15,16]. A recent study has shown that single item measures of motivation and ability are powerful predictors of behavior [17]. This provides the opportunity to use only two survey questions to identify drivers of behavior among Nigerian HCWs.

The following hypotheses are tested in this study:(1)HCWs with high motivation AND high ability are more likely to be fully vaccinated than HCWs with low motivation AND low ability.(2)HCWs with high motivation AND low ability as well as HCWs with low motivation AND high ability are more likely to be fully vaccinated than HCWs with low motivation AND low ability.(3)HCWs with high motivation AND high ability are more likely to be fully vaccinated than HCWs with high motivation AND low ability as well as HCWs with low motivation AND high ability.

## 2. Materials and Methods

### 2.1. Survey Data

Survey respondents were recruited via advertisements on Facebook using an open-source tool called Virtual Lab to optimize recruitment [18]. Recruitment was stratified by gender, age, geopolitical zone, education, and occupation. Twenty-three different ad sets were created on Facebook with distinct targeting and features aimed at recruiting individuals from each population of interest.

Survey respondents were offered a chance to win 4000 Naira (about USD 9.7) in mobile credit for answering the survey, with 1 in every 100 respondents winning the prize. The payments were made immediately upon completing the survey using Virtual Lab and Reloadly. This incentive was advertised directly in the ad: “Are you a healthcare professional? Take this quick survey for a chance to win N4000 in mobile credit”. 

Recruitment ads were targeted using Facebook’s category for industry of occupation, “Healthcare and Medical Services”. Demographic targeting was used to target individual strata except for occupation. To target workers from specific occupations (take “doctor” as an example), we generated creative text that started with “Are you a doctor?”, and the optimization routine boosted the budget on those ads in order to recruit more doctors. The same process was repeated for each occupation.

The campaign was seen by 64,320 people out of whom 2364 clicked on the ad, 697 gave consent and started the survey, and 496 completed it.

### 2.2. Operationalization of the Fogg Behavior Model: Motivation and Ability

Two questions were asked in the survey to operationalize the motivation and ability constructs. Responses to the question on motivation, “How important do you think getting the COVID-19 vaccine is for your health?”, were recorded on a Likert scale, which went from “not at all important” to “very important”. A binary variable was created for motivation, with HCWs who reported that it was very important for them to get a COVID-19 vaccine coded as 1 and other HCWs coded as 0. Responses to the question on ability, “How easy or difficult is it to get the COVID-19 vaccination for yourself? Would you say it is...?”, went from very difficult to very easy. A binary variable was also created for ability, with HCWs who felt that it would be very easy for them to get a COVID-19 vaccine coded as 1 and other HCWs coded as 0.

The FBM posits that motivation and ability need to be present for behavior to occur. A cross tabulation between the motivation and ability binary variables provides 4 possible combinations of motivation and ability: high motivation AND high ability, high motivation AND low ability, low motivation AND high ability, and low motivation and low ability. Previous studies have found 3 distinct levels of effects: (1) high motivation AND high ability; (2) high motivation AND low ability OR low motivation AND high ability; (3) low motivation AND low ability [13,17]. The hypotheses we tested in this study are consistent with these 3 levels of effects.

### 2.3. Completion of Two Doses of COVID-19 Vaccination

A question was asked in the survey to determine whether respondents had completed two doses of COVID-19 vaccination: “Have you received a COVID-19 vaccine?” Responses were recorded as “No”, “Yes, first dose only”, or “Yes, both doses”. A binary variable was created to measure whether HCWs had gotten two doses of a COVID-19 vaccine.

### 2.4. Statistical Analysis

Frequency distributions of the socioeconomic and demographic characteristics of HCWs, and their motivation and ability are shown in Table 1. Crosstabulations between HCW characteristics and having received two doses of a COVID-19 vaccine are also shown in Table 1.

The 3 hypotheses tested in this study are discussed in a previous section. Multivariate analysis was conducted to test these hypotheses, as shown in Table 2. Variables were introduced in stages: Socioeconomic and demographic characteristics were introduced in Model 1; the 4-category motivation–ability variable was introduced in Model 2. Statistical tests used in the analysis were considered significant at *p* < 0.05.

## 3. Results

Figure 2 shows the percentage of Nigerian HCWs who got two doses of a COVID-19 vaccination, the percentage who consider a COVID-19 vaccination to be very important for their health, and the percentage who find it very easy to get a COVID-19 vaccination for themselves. About one-third (33%) of HCWs had gotten two doses of a COVID-19 vaccine. While over two-thirds (69%) considered it very important to get a COVID-19 vaccination, less than one-third (32%) of HCWs found it very easy to get a COVID-19 vaccination for themselves.

Table 1, Column 1, shows the distribution of characteristics of the survey sample. About one-fifth of HCWs in the sample were from the North Central region, and more than one-quarter (29%) were from the South West. The largest city of Nigeria, Lagos, comprises the South West. Both the South South and South East regions comprised 15% of the sample. The North West and North East regions comprised 11% and 12% of the sample, respectively. Nearly half (49%) of HCWs were women. About 41% of HCWs had bachelor’s level education, 29% had a diploma or one year of full-time graduate study after the bachelor’s degree, and 13% had master’s or higher level education. One-quarter of the sample consisted of nurses or midwives, while physicians and community health workers comprised 16% and 12% of the sample, respectively.

About 43% of HCWs thought that the National Primary Heath Care Development Agency (NPHCDA) was definitely managing COVID-19 well, while 38% thought that the NPHCDA was managing the situation somewhat well. In terms of motivation and ability, 26% of respondents fell in the low motivation AND low ability segment, 47% fell in the high motivation OR high ability segment, and 27% fell in the high motivation AND high ability segment.

Columns 2 and 3 of Table 1 show the percentage of HCWs who reported that they had gotten two doses of a COVID-19 vaccine. Overall, 33% of HCWs reported that they had gotten two doses of a COVID-19 vaccine. About 44% of HCWs in the North West were fully vaccinated compared to 19% of respondents in the South South (*p* = 0.057), but the difference did not reach statistical significance at *p* < 0.05. About 43% of HCWs ages 40 and older compared with 30% ages 18–29 were fully vaccinated (*p* = 0.084). Being fully vaccinated increased with the level of education: 18% of HCWs with up to secondary education, 34% with bachelor’s, and 42% with master’s or higher were fully vaccinated (*p* = 0.010). Vaccination status varied by type of HCW: 51% of physicians, 34% of nurses or midwives, 28% of pharmacists, and 24% of patent and proprietary medicine vendors (PPMVs) had gotten two doses of a COVID-19 vaccine (*p* = 0.008). In Nigeria, owner-operated drug retail outlets, known as patent and proprietary medicine vendors (PPMVs), are a main source of medicines for acute conditions [19]. A PPMV is defined as “a person without formal training in pharmacy who sells orthodox pharmaceutical products on a retail basis for profit” [20]. About 36% of HCWs who felt that the National Primary Heath Care Development Agency (NPHCDA) was managing COVID-19 well were fully vaccinated, compared with 25% who reported that the NPHCDA was either not managing the COVID-19 situation well or that they did not know how well the NPHCDA was managing COVID-19, but the difference did not reach statistical significance.

There was a statistically significant relationship between the motivation and ability variable and being fully vaccinated: 64% of HCWs with high motivation AND high ability had gotten two doses of a COVID-19 vaccination, compared with 35% of HCWs with low motivation AND high ability, 27% of HCWs with high motivation AND low ability, and 11% of HCWs with low motivation AND low ability (*p* < 0.001).

Table 2 shows the adjusted odds of an HCW getting two doses of a COVID-19 vaccine. Model 1 in Table 2 shows that, compared with the South South, HCWs in the North West (aOR = 2.57, *p* < 0.05), North Central (aOR = 2.34, *p* < 0.05), and South East (aOR = 2.34, *p* < 0.05) were more likely to get two doses of a COVID-19 vaccine. Compared with HCWs who had up to secondary education, HCWs with bachelor’s (aOR = 2.55, *p* < 0.01), diploma (aOR = 3.09, *p* < 0.01), or master’s or higher education (aOR = 3.06, *p* < 0.01) were more likely to be fully vaccinated. Compared with nurses or midwives, physicians were more like to be fully vaccinated (aOR = 2.07, *p* < 0.05). HCWs who felt that the NPHCDA was definitely managing COVID-19 well were more likely to be fully vaccinated (aOR = 1.88, *p* < 0.05). There was no statistically significant relationship between age or gender and being fully vaccinated. Model 1 explains 7.61% of the variance in the likelihood of getting two doses of a COVID-19 vaccine.

Model 2 in Table 2 shows the odds of an HCW being fully vaccinated, after adjusting for the motivation and ability variable. Statistically significant regional differences in the likelihood of being fully vaccinated disappeared after adjusting for motivation and ability. This suggests that the lower likelihood of being fully vaccinated in the South South region is because of lower levels of motivation and ability to adopt a COVID-19 vaccination in the South South region. Although it became slightly weaker, the effect of education on being fully vaccinated remained statistically significant after adjusting for motivation and ability. The effect of being a physician became stronger after adjusting for motivation and ability (aOR = 3.11, *p* < 0.01). This suggests that had motivation and ability been higher among physicians, the likelihood of a physician being fully vaccinated would have been greater.

Finally, Model 2 in Table 2 shows that even after adjusting for region, age, gender, level of education, type of HCW, and trust in management of COVID-19 by the NPHCDA, there is a powerful relationship between motivation and ability and being fully vaccinated: HCWs with high motivation AND high ability had a 15-times higher odds of being fully vaccinated (aOR = 14.80, *p* < 0.001) compared with HCWs who had low motivation AND low ability. HCWs with low motivation AND high ability had a 4-times higher odds of being fully vaccinated (aOR = 4.25, *p* < 0.01) and HCWs with high motivation AND low ability had a 3-times higher odds of being fully vaccinated (aOR = 2.77, *p* < 0.01), compared with HCWs who had low motivation AND low ability.

Model 2 provides formal tests of Hypotheses 1 and 2. We also tested Hypothesis 3, that HCWs with high motivation AND high ability are more likely to be fully vaccinated than HCWs with either low motivation AND high ability or HCWs with high motivation AND low ability, by changing the reference category for the motivation and ability variable to each of these two categories in turn (not shown). We found that, after adjusting for other variables, HCWs with high motivation AND high ability had a 5-times higher odds ratio of being fully vaccinated compared with HCWs with high motivation AND low ability (aOR = 5.35, *p* < 0.001). We also found that HCWs with high motivation AND high ability had a nearly 4-times higher odds ratio of being fully vaccinated compared with HCWs with low motivation AND high ability (aOR = 3.48, *p* < 0.05). Consistent with previous studies [13,17], there was no statistically significant difference between the effect of high motivation AND low ability and the effect of low motivation AND high ability on being fully vaccinated. As mentioned, previous studies have found three levels of effects: (1) high motivation AND high ability; (2) high motivation AND low ability OR low motivation AND high ability; (3) low motivation AND low ability.

Figure 3 shows the distribution of motivation and ability by type of HCW in Nigeria. Overall, there was no significant difference in the distribution of motivation and ability among HCWs (*p* = 0.385). It is interesting to note, however, that the distribution of the motivation and ability variable among physicians was similar to the distribution of this variable among pharmacists: among physicians, 25% had high motivation AND high ability, 38% had high motivation AND low ability, and 30% had low motivation AND low ability; among pharmacists, 26% had high motivation AND high ability, 35% had high motivation and low ability, and 32% had low motivation AND low ability. This suggests that a physician’s higher odds ratio of being fully vaccinated is not because of their having higher motivation and ability than a pharmacist. Rather, there may be unobserved supply-side factors or a greater emphasis at the workplace for physicians to get vaccinated, which may contribute to higher levels of vaccination among physicians compared with other HCWs.

Figure 4 shows the percentage of HCWs who report that it is very easy for them to get a COVID-19 vaccine, by region. About 42% of HCWs in the North East and 37% in the North West reported that it was very easy for them to get a COVID-19 vaccination. By comparison, 24% of HCWs in the South South and 26% in the South East reported that it was very easy for them to get a vaccine. However, these differences did not reach statistical significance (*p* = 0.218).

## 4. Discussion

A successful COVID-19 pandemic response will depend upon the availability of a fully vaccinated workforce that can respond to the crisis. By being vaccinated, HCWs will not only be able to protect themselves and their families and feel confident in their interactions with patients, but they will also be able to advise their clients to get vaccinated against COVID-19 more persuasively. In Nigeria, by July 2021, government data suggest that only 20% of HCWs had been fully vaccinated [21]. An important constraint to HCW vaccination in Nigeria has been the limited availability of COVID-19 vaccines. How quickly the Nigerian healthcare workforce is vaccinated, as larger volumes of COVID-19 vaccines become available, is going to be critical to Nigeria’s pandemic response.

This is perhaps one of the first studies that uses a behavior model to examine drivers of COVID-19 vaccine uptake among HCWs. A behavior model that identifies the drivers of vaccine uptake can help inform the design of interventions that help HCWs overcome barriers to getting COVID-19 vaccinations. We found a powerful relationship between motivation and ability variable and the likelihood of getting two doses of a COVID-19 vaccine. Compared with HCWs with low motivation AND low ability, HCWs with high motivation AND high ability had a 15-times higher odds ratio of being fully vaccinated. Although there was a strong relationship between high motivation AND high ability and being fully vaccinated, the proportion of HCWs that fell in this segment was relatively low: Only 27% of HCWs had high motivation AND high ability.

The relatively low proportion of HCWs who had high motivation AND high ability was because of low ability among HCWs: Only 32% of HCWs reported that it was very easy to get a COVID-19 vaccination for themselves. Ability was particularly low in the South South (24%), and South East (26%) regions. Even in the regions where ability was higher, it did not reach 50%. These findings highlight the urgency of making COVID-19 vaccinations much easier for HCWs to get.

While there was no significant difference in the distribution of motivation and ability among different HCWs, physicians had a significantly higher odds of being fully vaccinated than other HCWs. These findings suggest that unobserved supply-side factors or a greater emphasis at the workplace for physicians to get vaccinated may be contributing to higher levels of vaccination among physicians compared with other HCWs. These findings may also suggest the effects of gender-related disparities—most Nigerian nurses and midwives are female, while most Nigerian physicians are male. In our study sample, about 84% of nurses and midwives were female, while 17% of physicians were female (not shown). This potential role of supply- and gender-related factors merits further examination.

What remains poorly understood is the factors that lower the ability of HCWs to get vaccinated. Anecdotal reports suggest that lack of childcare, the cost of transportation, and the inability to take time off from work may be significant barriers for HCWs to overcome [22]. HCWs may not be able to afford losing the time needed to recover from vaccine side effects. Some HCWs may be caregivers of adult family members, for example, and may have caregiving responsibilities with no one to substitute for them. The extent to which vaccines are available may also influence an individual’s ability to get vaccinated. A better understanding of the challenges faced by HCWs who may have multiple, overlapping, social and economic responsibilities is needed to design interventions that increase their ability to get a COVID-19 vaccination. Low ability is likely to keep motivation low. As ability increases, motivation may be expected to increase [16].

Social science theories of health behavior have tended to focus on motivation as the primary driver of behavior [14,15,16], with the result that ability factors have not received sufficient attention. There is a risk of underestimating the influence of practical barriers to vaccine adoption, such as those highlighted by the FBM: time, money, physical effort, mental effort, routine, and social norms. Our findings highlight the importance of paying greater attention to ability barriers that impede COVID-19 vaccine adoption among HCWs.

It is important to note that while HCWs with high motivation AND high ability are much more likely to be fully vaccinated (at 64%), there is still room to strengthen the relationship between high motivation AND high ability and being fully vaccinated: We need to understand how HCWs with high motivation AND high ability could come closer to being 100% vaccinated.

One limitation of this study is the reliance on self-reported ability, motivation, and vaccine uptake. Each of these may be influenced by social desirability or recall bias [23]. Another limitation is that, because of the cross-sectional nature of this study, no causal inferences can be drawn from the analysis.

The magnitude of the effects of motivation and ability on vaccination uptake is consistent with findings observed in studies examining the adoption of other preventive health behaviors [13,14,17] and suggests that the FBM provides a robust framework for studying vaccine uptake among HCWs in an LMIC. The FBM provides a practitioner-friendly framework for measuring changes in the drivers of behavior using a limited number of variables. This is advantageous in LMIC contexts where practitioners have limited technical and financial resources at their disposal and in situations where short turn-around times are needed to measure programmatic effects. The FBM could also be used for conducting real-time segmentation during implementation of community-based interventions to increase COVID-19 vaccine uptake. It would allow community health workers going door to door to use only two questions and a simple algorithm to determine which segment a particular member of the community falls in and intervene accordingly.

There is a need to collect evidence on which interventions increase motivation and ability of HCWs to get vaccinated. Until very recently, HCWs were largely ignored in the public health literature as recipients of vaccination. With the exception of a handful of studies conducted in HICs, little is known about drivers of vaccine uptake among HCWs.

Finally, as demonstrated by this study, online surveys can be a useful tool for measuring HCW vaccination uptake and its drivers. However, the extent to which online surveys represent the population of HCWs in Nigeria is not known. There is an urgent need to create a registry of HCWs in Nigeria, which can provide the sample frame for drawing a representative sample of HCWs. Only a sample drawn from such a sample frame can ensure that different types of HCWs or HCWs from different regions within Nigeria are well represented in the survey.

## 5. Conclusions

This is one of the first studies to examine the uptake of COVID-19 vaccines in an LMIC. A number of studies conducted in LMICs have examined the general population’s willingness to accept COVID-19 vaccines but it remains unclear whether this willingness will translate into the actual adoption of vaccines. We do not know, for example, what factors will impede the adoption of COVID-19 vaccines once they become available. Widespread availability of COVID-19 vaccines is expected in Nigeria and in many countries in sub-Saharan Africa by the last quarter of 2021.

This study examined vaccine uptake among Nigerian HCWs, ages 18 and older, who responded to an online survey conducted in July 2021. About one-third of HCWs in our sample reported having gotten two doses of a COVID-19 vaccine. Motivation and ability were powerful predictors of vaccine uptake, with HCWs who had high motivation and high ability having a 15-times higher odds ratio of being fully vaccinated. However, only 27% of HCWs had high motivation and high ability. About 5% of HCWs had high ability but low motivation. The remainder, 68% of HCWs, reported low ability to get vaccinated. This is remarkable given that HCWs were among the first populations prioritized for COVID-19 vaccination in Nigeria. If the majority of Nigerian HCWs do not find it easy to get a COVID-19 vaccine—at a time when they are being prioritized for vaccination—this needs the urgent attention of program designers and implementers.

Although we are not aware of any specific study that has examined the challenges that Nigerian HCWs face in getting vaccinated, anecdotal information suggests that the inability to take time off from work, the cost of transportation, elderly or childcare responsibilities, the physical effort required to visit a vaccination site, and adult vaccination not being the norm, may pose barriers to vaccine adoption. In contrast to their low level of ability to get vaccinated, HCWs’ motivation to get vaccinated was relatively high, with 69% of HCWs reporting that they were very motivated to get a COVID-19 vaccine for themselves. Current models of behavior tend to emphasize the importance of motivation while not paying sufficient attention to ability. In low resource settings, ability factors may play a critical role in ensuring that motivation is translated into behavior. Our findings suggest the need to use models of behavior that enable program designers and implementers to pay adequate attention to ability-related barriers to vaccine adoption.

## Figures and Tables

**Figure 1 vaccines-09-01162-f001:**
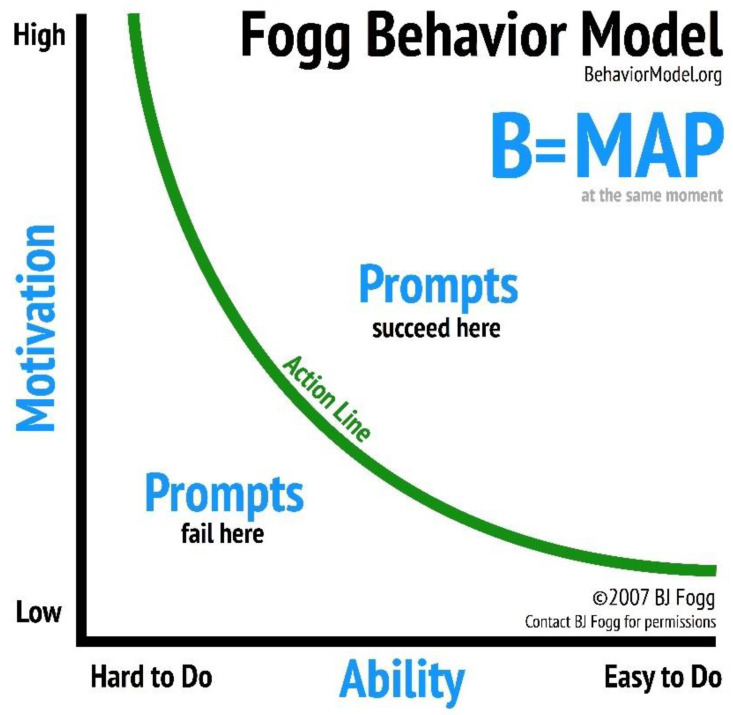
Fogg Behavior Model.

**Figure 2 vaccines-09-01162-f002:**
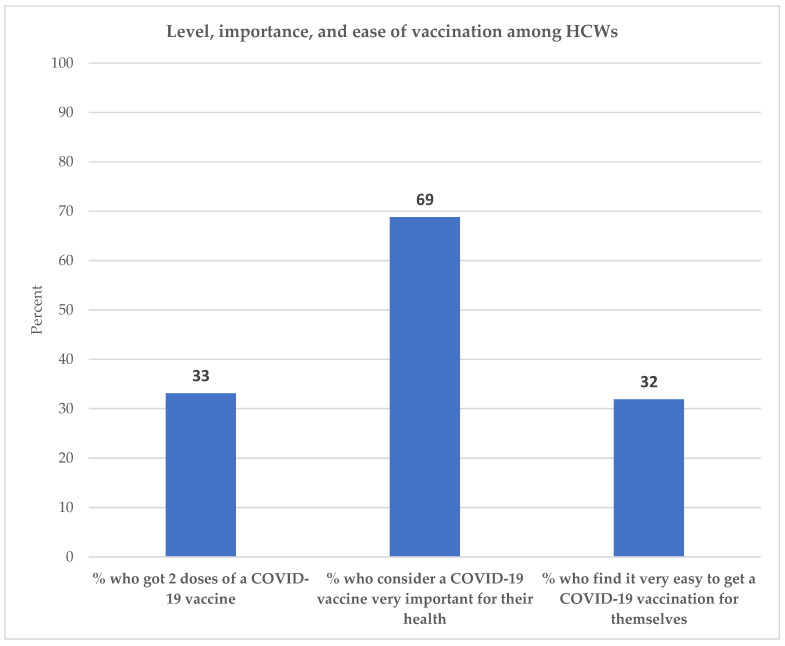
Percent of HCWs who got two doses of vaccine, consider vaccines important, and find them easy to get.

**Figure 3 vaccines-09-01162-f003:**
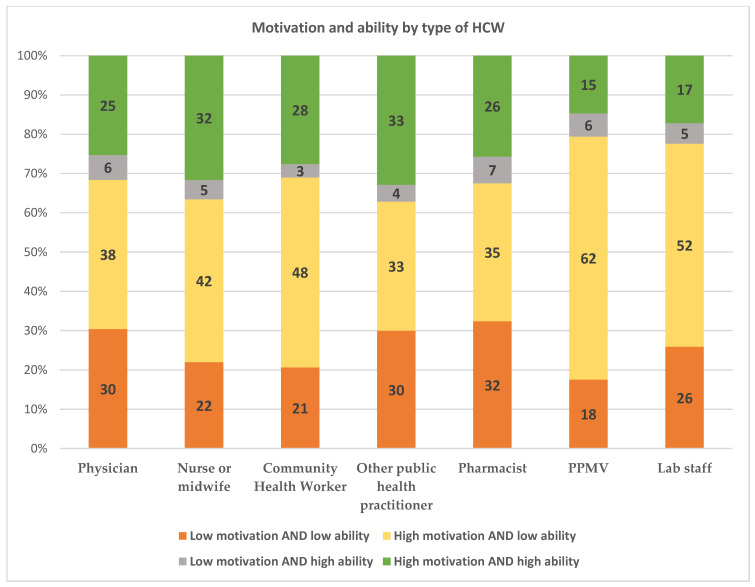
Distribution of motivation and ability by type of HCW.

**Figure 4 vaccines-09-01162-f004:**
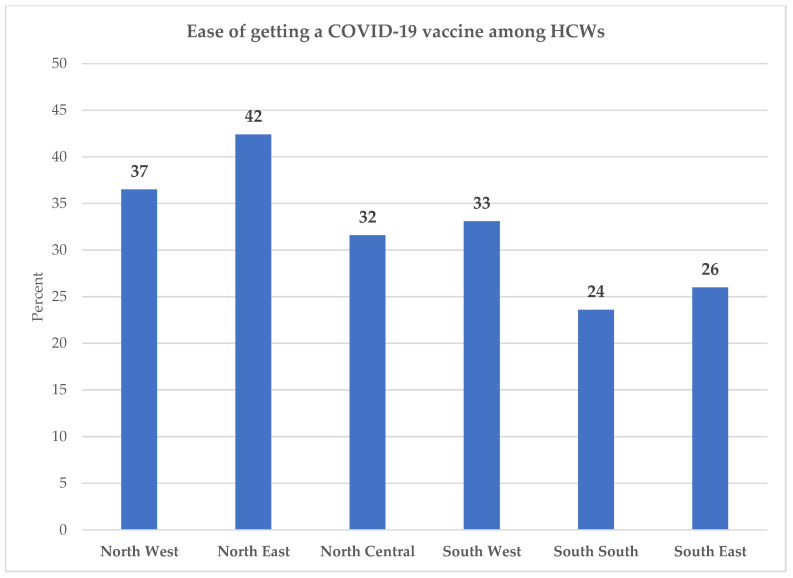
Percent of HCWs who find it very easy to get a COVID-19 vaccine by region.

**Table 1 vaccines-09-01162-t001:** Distribution of sample characteristics and % of HCWs who got two doses of a COVID-19 vaccine.

	(1) Distribution of Sample Characteristics	Percentage of HCWs Who Got Two Doses of a COVID-19 Vaccine		
		(2) % Other	(3) % Received Two Doses	(4) *p*-Value
Region				
North West	10.5% (*n* = 52)	55.8%	44.2%	
North East	11.9% (*n* = 59)	69.5%	30.5%	
North Central	19.8% (*n* = 98)	62.2%	37.8%	
South West	28.8% (*n* = 142)	68.3%	31.7%	
South South	14.5% (*n* = 72)	80.6%	19.4%	
South East	14.7% (*n* = 73)	63.0%	37.0%	0.057
Age				
18–29	30.4% (*n* = 151)	70.2%	29.8%	
30–39	51.2% (*n* = 254)	68.5%	31.5%	
40 and older	18.3% (*n* = 91)	57.1%	42.9%	0.084
Gender				
Male	51.2% (*n* = 249)	65.1%	34.9%	
Female	48.8% (*n* = 237)	68.8%	31.2%	0.384
Level of education				
Up to secondary	16.6% (*n* = 82)	81.7%	18.3%	
Bachelor’s	41.3% (*n* = 204)	66.2%	33.8%	
Diploma	28.9% (*n* = 143)	62.9%	37.1%	
Master’s or higher	13.2% (*n* = 65)	58.5%	41.5%	0.010
Type of healthcare workers (HCWs)				
Physician	15.9% (*n* = 79)	49.4%	50.6%	
Nurse of midwife	24.8% (*n* = 123)	65.9%	34.1%	
Community health worker	11.7% (*n* = 58)	69.0%	31.0%	
Other public health practitioner	14.1% (*n* = 70)	67.1%	32.9%	
Pharmacist	14.9% (*n* = 74)	71.6%	28.4%	
PPMV	6.9% (*n* = 34)	76.5%	23.5%	
Laboratory staff	11.7% (*n* = 58)	79.3%	20.7%	0.008
Is NPHCDA managing COVID-19 well?				
Not at all or don’t know	19.2% (*n* = 95)	74.7%	25.3%	
Yes, somewhat	38.3% (*n* = 190)	66.3%	33.7%	
Yes, definitely	42.5% (*n* = 211)	64.0%	36.0%	0.176
Motivation and ability				
Low motivation AND low ability	26.0% (*n* = 129)	89.1%	10.9%	
High motivation AND low ability	42.1% (*n* = 209)	73.2%	26.8%	
Low motivation AND high ability	5.2% (*n* = 26)	65.4%	34.6%	
High motivation AND high ability	26.6% (*n* = 132)	35.6%	64.4%	<0.001
Total	100% (*n* = 496)	66.9%	33.1%	

**Table 2 vaccines-09-01162-t002:** Logistic regression showing adjusted odds of an HCW getting two doses of a COVID-19 vaccination.

	Model 1 aOR (95% CI)	Model 2 aOR (95% CI)
Region		
North West	2.57 * (1.11–5.95)	2.25 (0.89–5.66)
North East	1.74 (0.75–4.03)	1.21 (0.48–3.06)
North Central	2.34 * (1.11–4.94)	2.17 (0.96–4.92)
South West	1.79 (0.87–3.68)	1.54 (0.70–3.41)
South South	1.00	1.00
South East	2.34 * (1.05–5.20)	2.30 (0.96–5.52)
Age		
18–29	1.00	1.00
30–39	1.04 (0.64–1.68)	1.04 (0.61–1.77)
40 and older	1.44 (0.79–2.63)	1.17 (0.59–2.29)
Gender		
Male	1.00	1.00
Female	0.94 (0.59–1.50)	1.08 (0.65–1.82)
Level of education		
Up to secondary	1.00	1.00
Bachelor’s	2.55 ** (1.27–5.12)	2.60 * (1.21–5.59)
Diploma	3.09 ** (1.52–6.26)	2.82 ** (1.30–6.12)
Master’s or higher	3.06 ** (1.32–7.04)	2.40 (0.95–6.07)
Type of healthcare worker (HCW)		
Physician	2.07 * (1.02–4.19)	3.08 ** (1.41–6.77)
Nurse of midwife	1.00	1.00
Community health worker	0.74 (0.35–1.53)	0.92 (0.42–2.05)
Other public health practitioner	0.89 (0.45–1.77)	0.98 (0.46–2.09)
Pharmacist	0.71 (0.36–1.43)	0.86 (0.40–1.84)
PPMV	0.71 (0.28–1.81)	0.94 (0.34–2.61)
Laboratory staff	0.45 (0.20–1.01)	0.65 (0.27–1.55)
Is NPHCDA managing COVID-19 well?		
Not at all or don’t know	1.00	1.00
Yes, somewhat	1.54 (0.83–2.83)	1.39 (0.71–2.72)
Yes, definitely	1.88 * (1.02–3.45)	1.29 (0.67–2.50)
Motivation and ability		
Low motivation AND low ability	Not included	1.00
High motivation AND low ability	Not included	2.77 ** (1.41–5.44)
Low motivation AND high ability	Not included	4.25 ** (1.42–12.69)
High motivation AND high ability	Not included	14.80 *** (7.26–30.19)
Pseudo R-squared	7.61%	20.33%
Number of respondents	484	484

* *p* < 0.05, ** *p* < 0.01, *** *p* < 0.001.

## Data Availability

The dataset used for the current study is available from the corresponding author on reasonable request.

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
