# Peer review of "Drivers of COVID-19 Vaccine Uptake amongst Healthcare Workers (HCWs) in Nigeria"

_vaccines, 2021, doi:10.3390/vaccines9101162_

Round 1
Reviewer 1 Report
Comment to the manuscript:
The authors aim was “to understand drivers of COVID-19 vaccination uptake among healthcare workers (HCWs) in Nigeria”.
The article tackles an essential aspect of the COVID-19 pandemic. However, several issues need revision.
Introduction
This section usually finishes with the aim or objectives of the study. This information is not specified in this section. In the method section, three hypotheses are proposed. This information may be moved to the introduction of the manuscript.
Abstract
In the abstract is written “HCWs with high motivation AND high ability” “AND” no capital letters required. The same applies to the word “OR”.
Method
Move the limitations of the study section to the discussion section of the manuscript. Regarding possible response bias, were any steps taken to reduce the risk of this type of bias?
The authors used Facebook to gather their data; “The campaign was seen by 64,320 people out of whom 2,364 clicked on the ad, 697 gave consent and started the survey, and 496 completed it”
What may be the consequences of the type of sampling method used?
What may be the consequences of a low response rate?
Results
The following senses are not clear: “Accordingly, after combining the category of high motivation AND low ability with the category of low motivation AND high ability, the 3-category variable described above was created”.
In statistical analysis section, indicate α (alpha, or p value) used to test hypothesis.
Table 1. Add the number of participants in each cell of the table. It would be adequate to distinguish the number of participants in each category of motivation and ability. In addition, to the combined categories presented in Table 1.
Improve figure titles. Ex: % (percentage)
The study focuses on getting two doses of the vaccine as an outcome variable; however, it may also be interesting to look at one vaccine dose participant’s characteristics and associations. According to the information provided in the method section, this data are available from the questionnaire applied in the study. Lines 213-215: “About 43% of HCWs ages 40 and older compared with 30% ages 18-29 were fully vaccinated (p=0.084) but the difference did not reach statistical significance at p<0.05”. Eliminate the last sentence, as shown.
To check the consistency of the responses would be adequate to know the percentage of personnel that indicated that it is "very easy" for them to get a COVID19 vaccine.
Line 225, eliminate the word “powerful”
Line 228, change p= 0.000 to p<0.001
Line 241, is it suitable to interpret the pseudo-R-squared as R-squared (coefficient of determination obtained in linear regression using continuous variables)?
Frequently, the results presented in the tables are repeated in the text. Only consider the most relevant to be described in the results section as text.
Discussion
A limitation of the study is the sampling technique and the low response rate. The authors should further discussion how this aspect impacted their results. Who does the sample represent? How offering a reward could affect the type of people who responded to the survey?
Was there any government policy (general or local level) that favor the physicians' vaccination over other types of health workers? Probably health authorities in Nigeria promote the vaccination of physicians, and maybe the results depend on health policy and not on individual decision.
Do all 10 authors fulfilled the criteria for authorship?
Author Response
REVIEWER 1
The authors aim was “to understand drivers of COVID-19 vaccination uptake among healthcare workers (HCWs) in Nigeria”.
The article tackles an essential aspect of the COVID-19 pandemic. However, several issues need revision.
Introduction
This section usually finishes with the aim or objectives of the study. This information is not specified in this section. In the method section, three hypotheses are proposed. This information may be moved to the introduction of the manuscript.
AS SUGGESTED, THIS INFORMATION HAS BEEN MOVED TO THE INTRODUCTION, JUST AFTER A DESCRIPTION OF THE FOGG BEHVIOR MODEL.
Abstract
In the abstract is written “HCWs with high motivation AND high ability” “AND” no capital letters required. The same applies to the word “OR”.
“AND” HAS BEEN PUT IN LOWER CASE IN THE ABSTRACT, AS SUGGESTED.
Method
Move the limitations of the study section to the discussion section of the manuscript. Regarding possible response bias, were any steps taken to reduce the risk of this type of bias?
STUDY LIMITATIONS HAVE BEEN MOVED TO THE DISCUSSION SECTION.
THE RESPONSE BIAS WAS REDUCED BY ENSURING COMPLETE ANONYMITY OF RESPONDENTS.
The authors used Facebook to gather their data; “The campaign was seen by 64,320 people out of whom 2,364 clicked on the ad, 697 gave consent and started the survey, and 496 completed it”
What may be the consequences of the type of sampling method used?
What may be the consequences of a low response rate?
IN ANY SURVEY, BUT ESPECIALLY WHEN SURVEYING REMOTELY OVER THE TELEPHONE OR THE INTERNET, MANY PEOPLE INVITED TO PARTICIPATE WILL CHOOSE NOT TO. THIS IMPLIES A SELECTION BIAS IN THAT THOSE WHO RESPOND WILL, BY DEFINITION, HAVE SOME UNOBSERVABLE TRAITS THAT MADE THEM RESPOND WHEN OTHERS DID NOT.
A FURTHER BIAS WHEN ADVERTISING ON SOCIAL MEDIA IS THAT SOME PEOPLE MIGHT BE MORE OR LESS LIKELY TO SEE THE AD OR BE SHOWN THE AD. THIS MAY DEPEND UPON HOW MUCH TIME THEY SPEND ON SOCIAL MEDIA AND HOW MANY OTHER COMPANIES ARE BIDDING FOR THEIR ATTENTION. AN IMMEDIATE CONSEQUENCE OF THESE BIASES IS THAT WE CANNOT ESTIMATE A POPULATION PARAMETER FOR "THE POPULATION OF ALL HEALTHCARE WORKERS IN NIGERIA" WITHOUT KNOWING MORE ABOUT THE SELECTION PROCESS.
IN THIS PARTICULAR STUDY, THE PRIMARY ESTIMAND OF INTEREST IS THE VACCINATION RATE CONDITIONAL ON SELF-REPORTED ABILITY AND MOTIVATION TO GET VACCINATED. WHILE THE MARGINAL DISTRIBUTION OF ABILITY AND MOTIVATION MIGHT BE DIFFERENT IN THE OVERALL HEALTHCARE WORKER POPULATION AS COMPARED TO OUR SAMPLE, THAT DOES NOT AFFECT OUR CONDITIONAL ESTIMATES. OUR CONDITIONAL ESTIMATES ARE, HOWEVER, IMPLICITLY CONDITIONED ON "BEING EXPOSED TO OUR FACEBOOK ADS AND RESPONDING TO THEM". THIS IS A POTENTIAL THREAT TO EXTERNAL VALIDITY GIVEN THAT THOSE WHO RESPOND TO ADS TO TAKE A SURVEY, WITH THE SAME REPORTED ABILITY AND MOTIVIATION TO GET VACCINATED, COULD HAVE VERY DIFFERENT VACCINATION RATES. IT DOES NOT, HOWEVER, POSE ANY THREAT TO THE INTERNAL VALIDITY OF THESE COMPARISONS GIVEN OUR POPULATION.
Results
The following senses are not clear: “Accordingly, after combining the category of high motivation AND low ability with the category of low motivation AND high ability, the 3-category variable described above was created”.
THE SENTENCE HAS BEEN REWRITTEN TO IMPROVE CLARITY.
In statistical analysis section, indicate α (alpha, or p value) used to test hypothesis.
THE STATEMENT “STATISTICAL TESTS USED IN THE ANALYSIS WERE CONSIDERED SIGNIFICANT AT P<0.05” HAS BEEN ADDED TO THE STATISTICAL ANALSYIS SECTION.
Table 1. Add the number of participants in each cell of the table. It would be adequate to distinguish the number of participants in each category of motivation and ability. In addition, to the combined categories presented in Table 1.
THE NUMBER OF PARTICIPANTS HAS BEEN ADDED TO TABLE 1.
Improve figure titles. Ex: % (percentage)
THIS IMPROVEMENT HAS BEEN MADE.
The study focuses on getting two doses of the vaccine as an outcome variable; however, it may also be interesting to look at one vaccine dose participant’s characteristics and associations. According to the information provided in the method section, this data are available from the questionnaire applied in the study.
IT WOULD, INDEED, BE VERY INTERESTING BUT IT MIGHT DISTRACT FROM THE CURRENT CLARITY OF THE PAPER. THIS CLARITY IS PARTICULARY IMPORTANT FOR PRACTITIONERS WHO HAVE NOT SEEN MUCH IN TERMS OF BEHAVIORAL SCIENCE APPLICATIONS TO IMMUNIZATION UPTAKE IN LMICs. HOWEVER, IF THE EDITORS FEEL THAT THIS IS IMPORTANT, WE CAN DO FURTHER ANALSYIS AND DIFFERENTIATE HCWS WHO GET 1 DOSE FROM HCWS WHO GET 2 DOSES.
Lines 213-215: “About 43% of HCWs ages 40 and older compared with 30% ages 18-29 were fully vaccinated (p=0.084) but the difference did not reach statistical significance at p<0.05”. Eliminate the last sentence, as shown.
THE LAST SENTENCE HAS BEEN ELIMINATED.
To check the consistency of the responses would be adequate to know the percentage of personnel that indicated that it is "very easy" for them to get a COVID19 vaccine.
NOT SURE IF WE UNDERSTOOD THIS COMMENT.
Line 225, eliminate the word “powerful”
POWERFUL HAS BEEN REMOVED FROM THIS SENTENCE.
Line 228, change p= 0.000 to p<0.001
DONE.
Line 241, is it suitable to interpret the pseudo-R-squared as R-squared (coefficient of determination obtained in linear regression using continuous variables)?
WE HAVE REMOVED THE SENTENCE IN WHICH THIS INTERPRETATION WAS PRESENTED.
Frequently, the results presented in the tables are repeated in the text. Only consider the most relevant to be described in the results section as text.
WE HAVE REMOVED SOME OF THE RESULTS THAT SEEMED LESS IMPORTANT FROM THE NARRATIVE.
Discussion
A limitation of the study is the sampling technique and the low response rate. The authors should further discussion how this aspect impacted their results. Who does the sample represent? How offering a reward could affect the type of people who responded to the survey?
WE HAVE RESPONDED TO THESE QUESTIONS ABOVE.
Was there any government policy (general or local level) that favor the physicians' vaccination over other types of health workers? Probably health authorities in Nigeria promote the vaccination of physicians, and maybe the results depend on health policy and not on individual decision.
WE ARE NOT AWARE OF ANY SUCH POLICY.
Do all 10 authors fulfilled the criteria for authorship?
WE HAVE REVIEWED THE LIST OF CO-AUTHORS AND REMOVED ONE OF THEM.
Reviewer 2 Report
In this study, Agha et al. investigate the drivers that predict a healthcare worker in Nigeria will get the COVID-19 vaccine using the Fogg Behavior Model and multivariate logistic regression analysis of survey responses. I read with interest the application of a behavioral model to better understand vaccine hesitancy among healthcare workers for the purpose of informing potential strategies for intervention. The findings of the study are important and timely. Overall, the study is well-conceived, well-written, and the experimental design is sound with the use of multivariate logistic regression models following adjustment of pre-defined confounders. My comments are minor and relate more to the presentation of the study than anything substantive.
(1) In the Introduction, the authors state the following twice: "Physicians were more likely to accept a vaccine than nurses..." (lines 47-48 and 51-52). Please remove redundancy and make the text more concise.
(2) For all figures in the manuscript (see Figs. 1, 2, 3, and 4), please label by placing the figure number and a descriptive title with or without a legend below the figure. For example, in Figure 1 remove Fogg Behavior Model and B=Map and place that information in the figure legend below the schematic of the model.
(3) Figure 1 -- please label the axes [e.g., Percent Respondents (y-axis) versus Survey Category (x-axis)]. Also, begin the figure title by spelling out %--it should be percentage. Figure legend should be placed below the figure as mentioned.
(4) Please clarify "different ad sets were created on Facebook with distinct targeting and creative to recruit individuals". "Creative" seems to be used as a noun in this instance. Please clarify its use in survey respondent recruitment.
(5) Line 201: "nursers" should be "nurses".
(6) Remove annotation for Table 1. You do not need "*p<0.05, **p<0.01, ***p<0.001". P values are already provided in the table.
Author Response
REVIEWER 2
In this study, Agha et al. investigate the drivers that predict a healthcare worker in Nigeria will get the COVID-19 vaccine using the Fogg Behavior Model and multivariate logistic regression analysis of survey responses. I read with interest the application of a behavioral model to better understand vaccine hesitancy among healthcare workers for the purpose of informing potential strategies for intervention. The findings of the study are important and timely. Overall, the study is well-conceived, well-written, and the experimental design is sound with the use of multivariate logistic regression models following adjustment of pre-defined confounders. My comments are minor and relate more to the presentation of the study than anything substantive.
(1) In the Introduction, the authors state the following twice: "Physicians were more likely to accept a vaccine than nurses..." (lines 47-48 and 51-52). Please remove redundancy and make the text more concise.
THE TEXT HAS BEEN MADE MORE CONCISE.
(2) For all figures in the manuscript (see Figs. 1, 2, 3, and 4), please label by placing the figure number and a descriptive title with or without a legend below the figure. For example, in Figure 1 remove Fogg Behavior Model and B=Map and place that information in the figure legend below the schematic of the model.
WE HAVE PLACED THE FIGURE NUMBER AND DESCRIPTIVE TITLE BELOW EACH FIGURE.
(3) Figure 1 -- please label the axes [e.g., Percent Respondents (y-axis) versus Survey Category (x-axis)]. Also, begin the figure title by spelling out %--it should be percentage. Figure legend should be placed below the figure as mentioned.
THIS HAS BEEN DONE.
(4) Please clarify "different ad sets were created on Facebook with distinct targeting and creative to recruit individuals". "Creative" seems to be used as a noun in this instance. Please clarify its use in survey respondent recruitment.
AS PER THE RECOMMENDATION OF REVIEWER 2, ABOVE, CREATIVE TO RECRUIT” HAS BEEN CHANGED TO “FEATURES AIMED AT RECRUITING”.
(5) Line 201: "nursers" should be "nurses".
THIS EDITORIAL CORRECTION HAS BEEN MADE.
(6) Remove annotation for Table 1. You do not need "*p<0.05, **p<0.01, ***p<0.001". P values are already provided in the table.
THE ANNOTATION HAS BEEN REMOVED.
Reviewer 3 Report
Please see attached file.

Author Response
REVIEWER 3
This is an interesting and well written manuscript. Attention to the following should improve it. In particular, the use of Likert scales and their conversion to binary variables requires additional discussion and clarification.
Minor comments:
- In the introduction, line 45, the name “DRC Congo” is redundant because C stands for Congo. Change it to “the Democratic Republic of the Congo (DRC)”.
THIS CHANGE HAS BEEN MADE.
- In the introduction, line 46, change: “accept a COVID-19 vaccine” to “undergo COVID-19 vaccination”.
THE RECOMMENDED CHANGE HAS BEEN MADE.
- In the introduction, line 56, change: “comprising of studies” to “comprised of studies”.
THIS CHANGE HAS BEEN MADE.
- In section 2.1, line 120, change: “and creative to recruit” to “features aimed at recruiting”.
DONE. THANK YOU!
- In section 2.1, line 130, change: “…we generated creative that started with the text…” to “…we generated creative text that started with…”.
DONE.
- In section 3, line 223, change: “relationship” to “difference”.
THIS CHANGE HAS BEEN MADE.
- In figure 3, note that CHW is an abbreviation for community health worker.
WE HAVE REPLACED THIS FIGURE WITH ONE THAT SPELLS OUT “COMMUNITY HEALTH WORKERS”. THANK YOU!
- In the discussion, line 304, change: “HCWS” to “HCWs”.
THIS CHANGE HAS BEEN MADE.
- In the discussion, line 322, change: “them with” to “for them”.
DONE. THANK YOU!
Major Comments:
- In the introduction, lines 75-76, you noted that 38% of nurses or midwives were willing to get a COVID-19 vaccination. Do nurses and midwives have similar training and education levels in Nigeria? If not, it would be useful to know the percentage for nurses only (if available) because nursing resistance to vaccination is a conundrum.
UNFORTUNATELY, THE DATA IS NOT AVAILABLE SEPARATELY FOR NURSES AND MIDWIVES.
- In the introduction, line 84, the phrase “less than secondary education” requires clarification. Please distinguish between lower secondary education or the second stage of basic education (e.g.,middle school [also known as intermediate school, junior high school, or lower secondary school] is an educational stage which exists in some countries, providing education between primary school and secondary school) and upper secondary education (in the United States upper secondary school typically consists of 3 or 4 years of high school. In other countries, upper secondary education can be structured quite differently). What does “secondary education” encompass in Nigeria?
SECONDARY SCHOOLS IN NIGERIA ARE EXPECTED TO COMPLY WITH THE FEDERAL GOVERNMENT’S POLICY: SIX YEARS OF ELEMENTARY SCHOOL IS FOLLOWED BY SIX YEARS OF SECONDARY SCHOOL.
- If someone wins 4000 Naira, it seems as though they should receive a “mobile payment” rather than “mobile credit”. If that is incorrect, briefly explain how the mobile credit works.
MOBILE CREDIT IS DELIVERED USING AN INTERNATIONAL SERVICE CALLED RELOADLY WHICH IS INTEGRATED INTO THE SURVEY PLATFORM, VIRTUAL LAB. WINNING RESPONDENTS ARE ASKED FOR THEIR PHONE NUMBER AND MOBILE PHONE OPERATOR. THEY THEN RECEIVE 4000 NAIRA IN THEIR ACCOUNT AUTOMATICALLY, WHICH THEY CAN SPEND ON ANY PRODUCT THEIR MOBILE PHONE PROVIDER OFFER. THESE ARE USUALLY PACKAGES OF EITHER DATA OR VOICE/SMS CREDITS.
- A Likert scale is typicallya five, seven, or nine-point agreement scale used to measure respondents' agreement with various statements. Likert scales are often used to allow the individual to express how much they agree or disagree with a particular statement.
- In order to judge whether converting your Likert scale(s) to binary variables was valid (and not negatively biased), all the potential responses for motivation and ability (not just the most negative and most positive) need to be available for review. This can be provided in the text or in a figure. I wonder if a format such as the one illustrated below
might have been more amenable to a binary variable analysis.
|
THE REVIEWER HAS RAISED AN EXCELLENT POINT.
A CONCERN WITH CONVERTING A VARIABLE ON A LIKERT SCALE TO A BINARY IS THAT THE RELATIONSHIP BETWEEN THE CONVERTED VARIABLE AND OTHER VARIABLES WILL CHANGE (MACCALUUM ET AL., 2002). WE TESTED WHETHER THIS HAPPENED USING THE NIGERIA DATA.
FIRST, WE LOOKED AT THE RELATIONSHIP BETWEEN THE TWO INDEPENDENT VARIABLES, MOTIVATION AND ABILITY. THE CORRELATION BETWEEN THESE TWO VARIABLES BEFORE DICHOTOMIZATION WAS r = 0.221 (r2 = 0.049). THE CORRELATION BETWEEN THESE TWO VARIABLES AFTER DICHOTOMIZATION WAS r=0.218 (r2 = 0.048).
SECOND, WE LOOKED AT THE RELATIONSHIP BETWEEN EACH OF THE INDEPENDENT VARIABLES AND THE OUTCOME, RECEIVING TWO DOSES OF A COVID-19 VACCINE. THE CORRELATION BETWEEN THE OUTCOME AND MOTIVATION WAS r = 0.334 (r2 = 0.112) BEFORE THE DICHOTOMIZATION. THE CORRELATION BETWEEN MOTIVATION AND THE OUTCOME AFTER DICHOTOMIZATION WAS r=0.343 (r2 = 0.118). BEFORE THE DICHOTOMIZATION, THE CORRELATION BETWEEN THE OUTCOME AND ABILITY WAS r = 0.506 (r2 = 0.256). THE CORRELATION BETWEEN ABILITY AND THE OUTCOME AFTER DICHOTOMIZATION WAS r=0.429 (r2 = 0.184).
IN ALL INSTANCES, ABOVE, THERE WAS NO CHANCE IN OUR SUBSTANTIVE FINDINGS AND INTERPRETATION REGARDING THE CONVERSION OF THE LIKERT SCALE VARIABLES TO DICHOTOMOUS VARIABLES.
THIS SUGGESTS THAT CONVERTING THE MOTIVATION AND ABILITY VARIABLES FROM THE LIKERT SCALE TO DICHOTOMOUS VARIABLES DID NOT CREATE A BIAS.
- The validity of a Likert scale attitude measurement can be compromised due to social desirability. This means that individuals may lie to put themselves in a positive light. You have acknowledged this in section 2.5. Offering anonymity on self-administered questionnaires should reduce social pressure, and thus may likewise reduce social desirability bias. Were you able to assure anonymity for your respondents?
COMPLETE ANONYMITY WAS ENSURED FOR RESPONDENTS.
- In section 3, line 218, change: “PPMVs” to “patent and proprietary medicine vendors (PPMVs). Please explain that: PPMVs are defined as “a person without formal training in pharmacy who sells orthodox pharmaceutical products on a retail basis for profit”. Please also note that: In Nigeria, owner-operated drug retail outlets, known as patent and proprietary medicine vendors (PPMVs), are a main source of medicines for acute conditions (see: HEALTH POLICY AND PLANNING; 19(3):177–182 and PLoS One. 2015; 10(1): e0117165).
PPMV DEFINITION HAS BEEN ADDED. CITATIONS HAVE BEEN ADDED. THANK YOU!
- While not synonymous, ability is closely related to availability and discussing how availability influences ability would be useful.
GREAT POINT. WE HAVE ADDED A BRIEF DISCUSION OF THIS.
- In addition, it would be useful to at least speculate about low ability influences motivation. I suspect that motivation influences ability, but to a lesser degree.
WE HAVE MENTIONED THE POSITIVE CORRELATION BETWEEN MOTIVATION AND ABILITY.
(AS SHOWN ABOVE, THERE IS A SIGNFICANT CORRELATION BETWEEN MOTIVATION AND ABILITY OF r = 0.221 (r2 = 0.049). HOWEVER, BECAUSE OF THE CROSS-SECTIONAL NATURE OF THE DATA, IT IS NOT POSSIBLE TO SAY ANYTING ABOUT THE DIRECTION OF THE EFFECT.)
Round 2
Reviewer 3 Report
Please see attached file.

Author Response
PLEASE FIND OUR RESPONSE TO THE REVIEWER'S COMMENTS IN CAPS, BELOW.
The manuscript is improved. Addressing the following should make it even better.
Minor comments:
- On line 32, delete the apostrophe after HCWs.
DELETED
- On lines 59-61, you note 7 US studies and report only 6 percentages. Please address this discrepancy.
THE 7TH PERCENTAGE HAS BEEN ADDED. THANK YOU FOR CATCHING THIS!
- On lines 77-79, it is difficult to understand how 55% and 48%, regardless of the ratio of participants in the two groups, can combine to result in 56%. Please clarify.
GREAT CATCH. WE WENT BACK TO THE ARTICLE. THE ARTICLE MISREPORTED ITS OWN FINDINGS. THE 55% AND 48% WERE CORRECT BUT THE AVERAGE WAS 52%, NOT 56% AS THE ARTICLE REPORTED IN THE ABSTRACT. THIS HAS BEEN CORRECTED IN OUR PAPER.
- On line 203, what does “29% had a diploma” mean? Please clarify. From what I have read “A diploma is a higher education program that provides advanced knowledge and practical skills learning in a specific career field. Diploma programs usually last for one- to two years. In Nigeria, “A diploma refers to a document awarded to students who successfully complete preset coursework related to a particular topic, field or industry. Diplomas are often awarded by academic institutions after a degree program is completed” (see: https://www.academiccourses.com › Diploma › Nigeria).
THE REVIEWER IS CORRECT. A BRIEF EXPLATION OF “DIPLOMA” HAS BEEN ADDED TO THE PAPER.
5.On line 204, change “comprised” to “was comprised”.
DONE.
- In table 2, the p-value 0.000 is awkward. Do you mean <0.0001? If so amend this accordingly.
WE HAVE MADE THIS CORRECTION IN TABLE 1.
- On lines 386 and 387, change: “frame” to framework”.
WE MEANT SAMPLE FRAME, WHICH MAY NOT HAVE BEEN CLEAR. WE HAVE CHANGED FRAME TO SAMPLE FRAME.
- On line 388, change: “represented” to either “well represented” or “thoroughly represented”.
THIS HAS BENE CHANGED.
Major comments:
- On line 291, “the type of medical education physicians receive” may influence Therefore, please change: “Rather, this maybe a result of the type of medical education physicians receive” to “The type of medical education physicians and pharmacists receive may influence their motivation”.
THIS CHANGES HAS BEEN MADE.
- Figure 3 indicates that 78% of nurses and midwives are in the High motivation OR high ability + High motivation AND high ability categories versus 69% of physicians. In addition, 32% of nurses and midwives are in the High motivation AND high ability category versus 25% of physicians. Yet, nurses and midwives are vaccinated less frequently than physicians. This suggests the possibility that low motivation trumps high ability or that low availability for nurses and midwives is masked within the High motivation OR high ability category. Low availability could relate to the significant barriers you have noted from reference 22. Assuming (without specific knowledge) that physicians are more likely to be male and that nurses and midwives are more likely to be female, this may be a gender-related discrepancy. These possibilities should be speculated upon within or after lines 338-355 and should improve the discussion section considerably.
THIS IS A VERY THOUGHTFUL POINT. WE HAVE INCLUDED IT IN THE DISCUSSION AFTER LINES 338. THANK YOU!
